# Prevalence and Correlates of Weight Stigma among Postpartum Women in China

**DOI:** 10.3390/ijerph192214692

**Published:** 2022-11-09

**Authors:** Mei Sun, Jiayuan Peng, Lisa Lommel

**Affiliations:** 1Xiangya School of Nursing, Central South University, Changsha 410017, China; 2Xiangya Center for Evidence-Based Practice & Healthcare Innovation: A Joanna Briggs Institute Affiliated Group, Central South University, Changsha 410083, China; 3School of Nursing, University of California San Francisco, San Francisco, CA 94115, USA

**Keywords:** postpartum women, weight stigma, psychological distress, perceived weight stigma, internalized weight stigma

## Abstract

Weight stigma is linked to a variety of psychological and health problems. In the postpartum period, women may be more likely to experience weight-based discrimination because of their changing social roles, weight, and the new functions their bodies fulfill. However, few studies have explored the issue of postpartum women’s weight stigma. Thus, to investigate the prevalence and correlates of weight stigma for postpartum women in China, we conducted a cross-sectional study of 507 postpartum women. Results showed that almost one quarter (21.1%) of postpartum women claimed to have experienced perceived weight stigma (PWS). Two thirds (66.1%) scored at the mean and 14.8% at the highest levels of weight bias internalization (WBI). During the postnatal period, the conditions of those most likely to report experiences of weight-based discrimination included low income [b = −0.203, *p* = 0.004], occupation as a worker [b = 0.921, *p* = 0.017] or farmer [b = 0.826, *p* = 0.033], stress [b = 0.044, *p* = 0.035], depression [b = 0.057, *p* = 0.021], and higher WBIS [b = 0.018, *p* = 0.002]. In addition, postpartum women who lived alone [*b* = 7.511, *p* = 0.048], were overweight or obese [*b* = 5.443, *p* = 0.000], and had higher PWS [*b* = 0.897, *p* = 0.004] and anxiety symptoms [*b* = 0.219, *p* = 0.011] had higher levels of internalized weight stigma. Findings from this study provide a foundation to better understand characteristics of postpartum women in China who are at risk for weight stigma.

## 1. Introduction

Weight stigma (WS), also called weight bias and weight discrimination, refers to the discrimination, stereotyping, and social exclusion based on one’s weight [1]. It can be classified into three types: experienced WS, perceived WS (PWS), and internalized WS (or weight-related self-stigma). Experienced stigma occurs when someone has experienced bias directed at them. The person may be aware that this bias happened due to their weight and as a result develop a perceived WS. When the person accepts and endorses weight stigma and believes that negative weight-based bias applies to themselves, internalized WS happens [2].

PWS has previously been used to refer to both experienced and perceived WS [2]. Rarely are perceived and experienced WS expressly differentiated. PWS could refer to both perceived and experienced WS, although weight-related self-stigma is different from the other two categories of WS (PWS and experienced stigma), according to a recent systematic review and meta-analysis [3]. In accordance with previous conclusions [2,3], we utilized PWS in our investigation to refer to both actual and perceived WS.

WS has become a serious public health issue. It has been reported that the prevalence of internalized WS and PWS in adults is as high as 44–57% (Saudi Arabia 46.4% [4], United States 44% [5] and 57% [6]) or as low as 4.6–11% (Germany 7.3% [7], United States 11% [8], and the United Kingdom 4.6% [9]). This suggests that WS is prevalent globally. More importantly, the prevalence is rapidly increasing. A longitudinal survey in 2008 showed that from 1995 to 2006, the rate of WS in the United States increased by 66%, and it increased more rapidly among women (55%) than in men (23%) [10]. In a recent systematic review, psychological distress was recognized as a result of WS, including lowered self-esteem, anxiety, and depression [11]. Moreover, the emotional toll of WS has been documented to lower adult quality of life [12]. Behavioral effects are another consequence of WS. Adults who are subjected to WS, for instance, are less inclined to exercise, have lower levels of self-efficacy, and are more likely to binge eat [13,14]. Physiological effects of WS have also been confirmed, particularly regarding cortisol levels. In the presence of WS, the physiological stress response is triggered, resulting in the release of cortisol, which then contributes to fat storage and unhealthy eating behaviors [15,16].

Natural weight fluctuations, societal views of an “ideal” postpartum body image, and the pressure to quickly “bounce back” after birth to a pre-pregnancy body image make the postpartum period a particularly unique and vulnerable time [17], and women are more susceptible to the effects of WS during this time. To make matters worse, the infant may also be impacted by maternal WS. WS may discourage women from breastfeeding [18]. It is also possible to influence cortisol levels in the child by affecting the mother’s cortisol secretion, promoting the same cortisol-related effects on eating and weight in the child [19].

To date, it appears that although postpartum women’s WS might be a common problem, it remains and underreported problem. Limited studies have examined the prevalence and average level of WS among postpartum women in the US. A US study of 358 postpartum women reported that almost one in five women (*n* = 63) reported experiencing varying degrees of WS by providers in the medical setting [18]. A prospective longitudinal study in the US involving 103 women showed that the mean Weight Bias Internalization Scale (WBIS) score for postpartum women was 26.86 ± 13.03 (Scores range from 11 to 77) [20].

To improve Chinese women’s postpartum health and prevent the detrimental effects of WS, a detailed understanding of WS is required. According to cultural customs, to promote the health of mothers and babies after childbirth and during lactation, women in China have a “confinement in childbirth” during puerperium. This includes moving less, eating fewer fruits and vegetables, and drinking brown sugar water and a variety of nutritious soups [21]. It is important that women have a large body for postpartum bodily functions such as breastfeeding, and body image is rarely considered [22]. However, in recent years, the social trend of “thinness equals beauty” has become increasingly popular in China. Recent studies show that Chinese women have a very negative view of their body and a high level of endorsement of the thin ideal [23]. Postpartum women in China, who are situated between the rapidly changing traditional and modernized worlds, may experience internal uncertainty and identity crises, which in turn may have a negative impact on them. Therefore, research is necessary to better understand the current status of weight stigma and associated factors among postpartum women in China.

The purpose of our study was three-fold: (1) investigate the prevalence of WS (including internalized WS and PWS) among postpartum women in China, (2) understand sociodemographic factors that put postpartum women at risk for WS, and (3) understand the relationship between psychological distress and WS among postpartum women in China.

## 2. Methods

### 2.1. Design and Participants

This is a cross-sectional study. A convenience sampling method was used to recruit participants from the waiting rooms of a maternal and child health hospital and four community health centers during postpartum visits from December 2021 to March 2022. Women who were met the inclusion criteria and willing to participate in this study completed the questionnaires in a demonstration room under the guidance of trained research staff. It took 10–15 min to complete a questionnaire, which was collected by the researcher on site.

For participants, the inclusion criteria were as follows: (1) ≥18 years; (2) within 12 months postpartum; (3) able to speak and read Chinese fluently and thus independently complete this survey; and (4) willing study participants. People with a history of significant mental or physical health disorders such as developmental delay, or cognitive disorders were excluded.

The planned sample size was determined based on the findings of comparable surveys conducted in other countries [20]. With a confidence level (1 − α = 0.95), permissible error (*δ =* 1.2) and response rate of the survey (95%), the required sample size was calculated to be 477 cases using PASS 11.0. A total of 530 postpartum women participated in the survey; 23 surveys were excluded due to incomplete questionnaires (more than 20% of the entire questionnaire incomplete) and missing important information such as height and weight. A total of 507 valid questionnaires were obtained.

### 2.2. Variables and Measures

#### 2.2.1. Sociodemographic Factors, and Anthropometric Information

An author-designed questionnaire was prepared to collect the participants’ sociodemographic and anthropometric data. Specifically, the sociodemographic questions include age, ethnicity, educational levels, profession, income, and living conditions (living alone, living with spouse/children, living with parents/grandparents, and other). Anthropometric questions include weight (kg) and height (cm). A portable height and weight meter was used to gauge the participants’ height and weight. Body mass index (BMI; kg/m^2^) was calculated based on height and weight, and participants’ weight status was then classified with Chinese classification as underweight (BMI < 18.5 kg/m^2^), normal weight (18.5 ≤ BMI < 24.0 kg/m^2^), overweight (24.0 ≤ BMI < 28.0 kg/m^2^), or obese (BMI ≥28 kg/m^2^) [24].

#### 2.2.2. Perceived WS Questionnaire (PWS)

The PWS is a self-reported questionnaire that tracks perceptions of experiences with weight-based stigmatization. The scale uses 10 dichotomous items with a total score of 10. Items are scored as “yes” or “no”, with a score of 1 for “yes” and a score of 0 for “no”. Participants answered either yes or no on whether they had the described experience and feelings within the last week. A higher total score on the PWS indicates a higher level of perceived WS. The Chinese version of the PWS has been validated with Chinese university students and its internal consistency is acceptable (Cronbach’s alpha = 0.84) [25]. In the present study, PWS also showed good internal consistency with Cronbach α equaling 0.80.

#### 2.2.3. Weight Bias Internalization Scale (WBIS)

The WBIS consists of 11 items with a Likert scale rating of 1 (strongly disagree) to 5 (strongly agree). After reverse coding two item scores (items 1 and 9), the sum of the 11 item scores—which can reach a maximum of 55—represents the degree of internalized weight bias; a higher score denotes a higher level of internalization. “I hate my overweight” is an example item for the WBIS. The English version of the WBIS has shown good internal consistency (Cronbach’s α = 0.90) [26]. The original English version of the WBIS was translated into Chinese and now uses “weight” to replace “overweight”. For example, the sample item above is “I hate my weight” in the Chinese WBIS. The Chinese version showed good internal consistency in a study examining the psychometric properties of the WBIS using samples of Chinese children and adolescents [27]. In the present study, WBIS showed good internal consistency with Cronbach α equaling 0.90. For WBIS scores, there are no specified thresholds. The mean in this study was 25.50 ± 9.26, and we utilized that as our cutoff point and considered those who scored more than that to have a high level of weight biased internalization.

#### 2.2.4. Depression Anxiety Stress Scale-21 (DASS-21)

The 21-item DASS-21 was used to measure maternal psychological status. Depression (7 items), anxiety (7 items), and stress (7 items) are three distinct types of psychological distress assessed by the DASS-21 [27]. The scale uses a four-point Likert scale to evaluate each item (0 being not at all, and 3 being very much). The sum of the item scores multiplied by two can be used to generate the three subscale scores, with larger scores denoting higher degrees of psychological distress. Depression cutoffs are 10–13 (mild), 14–20 (moderate), 21–27 (severe), and over 28 (extremely severe); anxiety cutoffs are 8–9 (mild), 10–14 (moderate), 15–19 (severe), and over 20 (extremely severe); and stress cutoffs are 15–18 (mild), 19–25 (moderate), 26–33 (severe), and over 34 (extremely severe) [28]. The DASS-21’s psychometric properties have been extensively researched and found to be robust [29]. The Chinese version of the DASS has good internal consistency indices (Cronbach’s alpha) of 0.83, 0.80, and 0.82 for the depression, anxiety, and stress subscales, respectively, and 0.92 for the total DASS total [30].

### 2.3. Statistical Analysis

SPSS 26.0 software (IBM SPSS Inc., Chicago, IL, USA) was used to analyze the data. Categorical variables in socio-demography, psychological distress, PWS, and internalized WS were summarized as frequency counts (percentages), where continuous variables were summarized as means, standard deviations, and ranges. According to the Kolmogorov–Smirnov criteria, the total scores of the PWS and WBIS were not normally distributed. Therefore, the Mann–Whitney U non-parametric test (for 2-level variables, such as marital status) or the Kruskal–Wallis non-parametric test (for variables with 3 or more levels, such as occupation) was used to determine the association between postpartum women’s sociodemographic characteristics and WS (including internalized WS and PWS). Then, Spearman’s analysis was used to examine the correlation between PWS, internalized WS, and psychological distress. Variables that were statistically significant in the bivariate analysis were included in the multivariate analysis. Multiple linear regression (with input regression variable selection) was used to explore the major factors affecting perceived and internalized WS among postpartum women. Level of significance was set at *p* < 0.05 (two-tailed).

### 2.4. Ethical Considerations

Prior to recruitment, ethical approval for this study was obtained from the Ethics Committee of the School of Nursing, Central South University (Approval no. E2021108). Before the investigation, participants were informed about this study, the voluntary and anonymous nature of participation, and that the study would not cause harm to participants. Verbal and written consent was obtained. All collected data was kept confidential, and only researchers had access to the database’s encrypted data.

## 3. Results

### 3.1. Sociodemographic Characteristics

On average, participants were 30.92 (SD = 4.67) years old. The mean BMI of the 507 participants was 22.94 ± 3.00. About 4.70% of the participants were low weight, 61.10% were normal weight, and 34.10% were overweight or obese (see Table 1).

### 3.2. Descriptive Analysis of PWS, Internalized WS

The mean scores for PWS and WBIS were 0.46 ± 1.23, 25.50 ± 9.26, respectively. In this study in China, the prevalence of maternal perceived WS was 21.1% (*n* = 107), and 66.1% (*n* = 335) and 14.8% (*n* = 75) of postpartum women endorsed the mean and highest levels of WBI, respectively (see Table 2).

### 3.3. Descriptive Analysis of Psychological Distress

The mean scores for depression, anxiety, and stress were 5.58 ± 8.18, 7.44 ± 8.13, and 9.20 ± 9.44, respectively. A small percentage (15.4%; *n* = 78) of postpartum women had mild/moderate depression, and slightly more (7.3%; *n* = 37) had severe/very severe depression. Approximately one fourth (22.7%; *n* = 115) had mild/moderate anxiety, while 15.6% (*n* = 79) had severe/very severe anxiety. A small percentage (15.2%; *n* = 77) had mild/moderate stress and 8.3% (*n* = 42) had severe/very severe stress (see Table 3).

### 3.4. Univariate Analyses of the Factors Associated with Internalized WS and PWS

The Mann–Whitney U test and Kruskal–Wallis test showed that postpartum women’s age (*p* = 0.006), living conditions (*p* = 0.025), and BMI (*p* = 0.000) were associated with internalized weight stigma. Postpartum women aged between 39 and 48 years, whose BMI indicated being overweight or obese, and who were living alone had higher levels of WBIS. Additionally, postpartum women’s occupation (*p* = 0.005), monthly per capita household income (*p* = 0.002), and living conditions (*p* = 0.001) were associated with PWS. PWS levels were higher among postpartum women who lived alone, had a per capita monthly household income of less than RMB1000, and whose occupations were workers or farmers (see Table 4).

### 3.5. Correlations between Internalized WS, PWS, and Psychological Stress

The Spearman correlation analysis showed significant correlations between WBIS, PWS, and DASS-21. Additionally, each relationship was going in the predicted direction. PWS (*r_s_* = 0.194, *p* < 0.01), depression (*r_s_* = 0.381, *p* < 0.01), anxiety (*r_s_* = 0.373, *p* < 0.01), and stress (*r_s_* = 0.363, *p* < 0.01) were positively associated with internalized WS. In addition, WBIS (*r_s_* = 0.194, *p* < 0.01), depression (*r_s_* = 0.335, *p* < 0.01), anxiety (*r_s_* = 0.328, *p* < 0.01), and stress (*r_s_* = 0.350, *p* < 0.01) were positively associated with perceived WS (see Table 5).

### 3.6. Regression Analysis of WS (Including Internalized WS and PWS)

Regression analyses of WS (including internalized WS and PWS) presented in Table 6 had adjusted R^2^ of 0.377 and 0.278, respectively; that is, approximately 38% and 28% of the scale change was accounted for by predictors selected by the input method from all significant bivariate associations. The conditions of postpartum women who internalized the worst WS included living alone [*b* = 7.675, *p* = 0.044], higher BMI [*b* = 5.460, *p* = 0.000], higher PWS [*b* = 0.847, *p* = 0.006], and anxiety [*b* = 0.434, *p* = 0.012]. The conditions of postpartum women with higher levels of perceived WS included low income [*b* = −0.203, *p* = 0.004], occupation as a worker [*b* = 0.921, *p* = 0.017] or farmer [*b* = 0.826, *p* = 0.033], stress [*b* = 0.044, *p* = 0.035], depression [*b* = 0.057, *p* = 0.021], and higher WBIS [*b* = 0.018, *p* = 0.002] (see Table 6).

Score range of continuous variables: PWS (0~10), WBIS (11–55), Depression (0–42), Anxiety (0–42), Stress (0–42), Age (19–48), BMI (9.57–34.45). Coding of categorical variables: Living conditions: Living alone = 0, living with spouse/children = 1, living with parents/grandparents = 2, Other = 4.

## 4. Discussion

Existing systematic reviews and original research focusing on WS among the general population have found that WS is related to a range of sociodemographic variables, including income and BMI category [31,32]. The present study shows that these sociodemographic factors are also relevant in postpartum women in China. Our study has added to the existing evidence that postpartum women’s WS-related factors in China include occupation (worker or farmer) and living conditions (living alone). These new discoveries help build the evidence base of knowledge on the development of WS among postpartum women in China and have practical implications for interventions to prevent or reduce WS among this population of postpartum women.

### 4.1. The Prevalence of WS and Correlates among Postpartum Women in China

Among participants in our study, 21.1% (*n* = 107) reported perceived discrimination due to weight, which is higher than the rates in the general population in the United States (11%) [8], England (4.6%) [9] and Germany (7%) [7]. This finding may be due to sample differences. Women are more likely to be affected by WS than men, with a 60% increased risk of experiencing WS [33]. However, studies in Germany, the United States and England have included both men and women in their samples, which may explain lower rates of perceived stigma [7,8,9]. Age is another cause of heterogeneity. An English study indicated the prevalence of WS declines with age [9]. “Health was more important to older adults than physical attractiveness” for older adults [34].

Furthermore, our findings show that internalized weight bias is prevalent among postpartum women in China, with approximately 66.1% of postpartum women endorsing the WBI average (25.50 ± 9.26). Additionally, high levels of WBI (corresponding to 1 SD above the WBIS mean) were endorsed by 14.8% in this sample. The mean WBIS in our sample was slightly lower compared to the WBI average score in a sample of 103 overweight or obese postpartum women in the United States (M = 26.86) [20]. This small difference between our study and the US study may be due to the inconsistent BMI categories of the included population. Our study included postpartum women in all BMI categories (range = 15.82–34.45 kg/m^2^), whereas the US study included only two BMI categories of obese or overweight (range = 25.4–62 kg/m^2^). A study conducted with 868 young adults in Saudi Arabia suggested that those with higher WBI levels were individuals with a higher BMI [35]. Although lower than the WBI average score in the above studies [20], the prevalence of a mean level WBI in our study (66.1%) was higher than the results of previous studies for all weight range populations (USA: 44–57%; Saudi Arabia: 46.4%) and higher than the prevalence observed in studies for overweight and obese groups (Saudi Arabia: 57%) compared to other adult populations [4,5,6,35]. However, the prevalence of high levels of WBI was lower in our study than one in the United States of 3,504 adults with and without obesity (20%) [5]. This suggests that although the prevalence of WBI among postpartum women in China is higher than Western countries, the percentage of high internalization is lower when compared to the general public in other countries.

Our study found that postpartum women who were low-income and occupationally employed as workers or farmers were more likely to report experiences of perceived WS. This is in line with the findings of a previous study reporting that perceived stigma due to weight was more common among those with lower income [32]. In China, workers and farmers have a low socioeconomic status [36,37]. According to labelling theory, perceptions of stigma and discrimination are most apparent among vulnerable groups with fewer social resources [38].

PWS also showed a significant and strong positive relationship with psychological distress (depression and stress) for postpartum women. Longitudinal evidence suggests that people in general who perceived WS not only have more stress [32], but also more depressive symptoms [39]. This relationship may be bidirectional. This “reverse effect” was tested in a longitudinal study in England of older adults that observed a small but significant indirect effect of depressive symptoms on perceived WS, suggesting that individuals with psychological distress are more likely to perceive weight discrimination [40].

In addition, our results also show a pattern of sociodemographic characteristics among postpartum women in China who internalize weight bias. Similar to previous studies [33,35,41,42], our study found BMI to be positively associated with WBI. WBIS scores were higher in overweight (M = 30.57) and obese (M = 31.75) postpartum women compared to those who were underweight (M = 15.83) or normal weight (M = 23.30). However, it is noteworthy that post hoc analyses revealed significant differences in WBI scores not only between normal weight postpartum women and overweight or obese postpartum women, but also between low weight postpartum women and postpartum women in the other three BMI categories. As a result, postpartum women across body weight categories may all be vulnerable to high WBI. These findings corroborate recent evidence of WBI observed among women in general of normal weight or underweight and underline the importance of studying WBI in populations with a variety of body weights [5]. In contrast to BMI, little research has examined the relationship between WBI and living conditions. In our study, levels of WBI were found to be lower in postpartum women who lived with their families and higher in those who lived alone. These differences might illustrate how protective social support is, as found in a longitudinal Canadian study of 3,388 pregnant women through one year postpartum [43]. Our study also explored the association between internalized stigma and psychiatric disorders among postpartum women. We found that internalized stigma was significantly associated with anxiety [b = 0.219, *p* = 0.011], which is in line with earlier studies in the general population that demonstrate a link between higher levels of internalization and the severe anxiety [13,35].

Internalized WS is a construct that is highly related to but distinct from perceived WS. Consistent with previous research, our findings suggest a significant positive correlation between WBI and PWS. A systematic review [3] suggests that perceived WS leads to greater internalized WS. This association between perceived and internalized stigma has also been confirmed in previous studies on self-stigma in a study of people with schizophrenia in Taiwan [44]: People initially encounter hostile behavior before realizing that the stigma is brought on by their traits (i.e., perceived stigma). Then, as a result of becoming aware of the stigma and developing self-stigma, individuals can embrace and support the discriminatory attitudes and behaviors they encounter. Our results confirm and support the hypothesis that perceived stigma may contribute to weight-related self-stigma.

### 4.2. Study Limitations

This study has several limitations. First, we evaluated WS using a self-reported scale, which could result in the hiding of subjective sensations. Second, the scope of the study population was constrained by the fact that it only included postpartum women residing in the Changsha region. Multicenter studies could be conducted in the future. Additionally, the prevalence of weight stigma and associated factors in men could be examined to see if similar results emerge.

### 4.3. Implications

This was the first study to initially explore WS and its associated factors among postpartum women in China. Current research suggests that WS is highly prevalent among postpartum women in China and is associated with negative psychology. This information will be valuable for women’s health in China, particularly in terms of mental health. Given recent calls from the international community for efforts to address WS [14], our findings add to the literature on the prevalence of weight stigma among postpartum women in China and add to the existing research evidence on associated factors. These new discoveries, as part of an effort to improve weight-related health globally, help build the evidence base of knowledge on the development of WS among postpartum women in China and have practical implications for interventions to prevent or reduce WS among this population of postpartum women.

## 5. Conclusions

Our study suggests a high prevalence of WS (including internalized WS and PWS) among postpartum women in China. Additionally, almost four-fifths of postpartum women in this study experienced moderate-to-high internalized WS. High internalized WS was associated with living alone, higher BMI, higher PWS, and anxiety. In addition, high perceived WS was associated with having a low income, an occupation as a worker or farmer, stress, depression, and high WBI. In the future, we need to conduct longitudinal studies to explore the causal relationships of these associated factors. Additionally, we need to develop targeted intervention programs to provide comprehensive interventions for the postpartum population in China to reduce the prevalence of WS and its negative impact on maternal outcomes.

## Figures and Tables

**Table 1 ijerph-19-14692-t001:** Sociodemographic Characteristics of the Participants (*n* = 507).

Variable	Category	*n*	%
Age (years)	19–28	161	31.8
	29–38	315	62.1
	39–48	31	6.1
Ethnicity	Han	469	92.5
	Minority	38	7.5
Educational levels	Primary	2	0.4
	Junior high school	18	3.6
	High School	51	10.1
	Undergraduate	358	70.6
	Graduate	78	15.4
Professions	Farmer	8	1.6
	Worker	8	1.6
	Staff	205	40.4
	Medical personnel	122	24.1
	Freelance	121	23.9
	Others ^a^	43	8.4
Monthly income (yuan)	<1000	8	1.6
	1000~3000	36	7.1
	3000~5000	119	23.5
	>5000	344	67.9
Living conditions	Living alone	6	1.2
	Living with spouse/children	390	76.9
	Living with parents/grandparents	101	19.9
	Other	10	2
BMI category	Underweight	24	4.7
	Normal weight	310	61.1
	Overweight	141	27.8
	Obese	32	6.3

Note: Abbreviations: BMI, Body Mass Index. ^a^ “Others” included teacher, housewife, programmer, or student.

**Table 2 ijerph-19-14692-t002:** Average Scores for Scales and Prevalence of Weight Stigma (*n* = 507).

Variables	Mean (SD)	Yes (%)	No (%)	Level of Severity (%)		
				Low Internalization	Mean Internalization	High Internalization
PWS	0.46 (1.23)	21.1 *	78.9 *	—	—	—
WBIS	25.50 (9.26)	—	—	19.1 *	66.1 *	14.8 *

Note: Abbreviations: PWS, Perceived Weight Stigma questionnaire; WBIS, Weight Bias Internalization Scale. Score range: PWS (0~10), WBIS (11~55). * PWS score ≥ 1 is Yes; equal to 0 is No. Low internalization (below the mean 1 SD) corresponded to WBIS scores ≤ 16.24, mean internalization corresponded to WBIS scores 16.25–34.75, and high internalization (above the mean 1 SD) corresponded to WBIS scores ≥ 34.76.

**Table 3 ijerph-19-14692-t003:** DASS-21 Dimensional Scores and Prevalence of Depression, Anxiety, and Stress (*n* = 507).

Variables	Mean (SD)	Level of Severity (%)				
		Normal	Mild	Moderate	Severe	Very Severe
Depression	5.58 (8.18)	77.3	8.1	7.3	3.7	3.6
Anxiety	7.44 (8.13)	61.7	5.7	17.0	6.5	9.1
Stress	9.20 (9.44)	76.5	7.9	7.3	5.3	3.0

Note: Abbreviations: DASS-21, Depression Anxiety Stress Scales.

**Table 4 ijerph-19-14692-t004:** Univariate Analysis of PWS and Internalized WS (*n* = 507).

Characteristics	Internalized Weight Stigma		PWS	
	Mean ± SD	H/Z (*p*)	Mean ± SD	H/Z (*p*)
Age (years)		10.386 * (0.006)		5.623 (0.060)
19–28	25.75 ± 8.92		0.41 ± 1.17	
29–38	24.86 ± 9.23		0.41 ± 1.09	
39–48	30.71 ± 9.82		1.23 ± 2.29	
Ethnicity		−0.973 (0.331)		−0.554 (0.579)
Han	25.35 ± 9.10		0.47 ± 1.27	
Minority	27.37 ± 10.98		0.32 ± 0.57	
Educational levels		3.187 (0.527)		8.439 (0.077)
Primary	26.50 ± 4.95		0.50 ± 0.71	
Junior high school	27.61 ± 11.70		1.50 ± 2.71	
High school	27.67 ± 10.38		0.69 ± 1.73	
Undergraduate	25.17 ± 8.99		0.37 ± 0.96	
Graduate	25.10 ± 9.12		0.49 ± 1.35	
Professions		4.898 (0.298)		14.999 (0.005)
Farmer	30.75 ± 11.74		1.50 ± 3.07	
Worker	26.63 ± 11.38		1.75 ± 2.32	
Staff	24.58 ± 8.25		0.26 ± 0.74	
Medical personnel	26.73 ± 9.86		0.44 ± 0.95	
Freelance	25.42 ± 8.90		0.60 ± 1.61	
Others ^a^	25.44 ± 11.69		0.60 ± 1.53	
Monthly income (yuan)		5.238 (0.155)		14.439 * (0.002)
<1000	31.00 ± 11.28		2.50 ± 3.51	
1000~3000	27.72 ± 9.59		0.97 ± 2.10	
3000~5000	24.50 ± 9.25		0.41 ± 1.05	
>5000	25.49 ± 9.13		0.37 ± 1.01	
Living conditions		9.323 * (0.025)		10.593 (0.014)
Living alone	38.33 ± 9.71		2.67 ± 4.08	
Living with spouse/children	25.22 ± 9.23		0.42 ± 1.18	
Living with parents/grandparents	25.90 ± 8.63		0.38 ± 0.90	
Other	24.60 ± 11.93		1.30 ± 1.57	
BMI category		99.228 * (0.000)		2.887 (0.409)
Underweight	15.83 ± 5.74		0.50 ± 1.02	
Normal weight	23.30 ± 8.10		0.40 ± 1.05	
Overweight	30.57 ± 8.63		0.45 ± 1.31	
Obese	31.75 ± 10.23		1.03 ± 2.22	

Note: Abbreviations: PWS, Perceived Weight Stigma questionnaire; BMI, Body Mass Index. ^a^ “Others” included teacher, housewife, programmer, and student. * Significant post hoc (Nemenyi) contracts exist in age, living conditions, BMI category, and monthly income (*p* < 0.05), indicating that the age group (39–48), living conditions (living alone), BMI (overweight and obesity), and monthly income (1000 yuan) have the highest WBIS scores. Monthly income has significant post hoc (Nemenyi) contracts, indicating that the income group (1000) has the highest PWS scores.

**Table 5 ijerph-19-14692-t005:** Correlation Coefficients Between Internalized WS, PWS, and Psychosocial Stress (*n* = 507).

	1	2	3	4	5
1. WBIS	1.000				
2. PWS	0.194 **	1.000			
3. Depression	0.381 **	0.335 **	1.000		
4. Anxiety	0.373 **	0.328 **	0.792 **	1.000	
5. Stress	0.363 **	0.350 **	0.806 **	0.806 **	1.000

Note: Abbreviations: PWS, Perceived Weight Stigma questionnaire; WBIS, Weight Bias Internalization Scale. ** *p* < 0.01.

**Table 6 ijerph-19-14692-t006:** Multiple Regression Analysis with Internalized Weight Stigma and PWS as Dependent Variables (N = 507).

Variables	Internalized WS ^a^				PWS ^b^			
	*b*	*se*	*b’*	*p*	*b*	*se*	*b’*	*p*
Age (years)	0.001	0.072	0.000	0.992	—	—	—	—
Living condition (other)	Reference							
Living condition (Living with parents/grandparents)	4.026	2.454	0.174	0.102	−0.496	0.354	−0.161	0.162
Living condition (Living with spouse/children)	3.528	2.375	0.161	0.138	−0.374	0.343	−0.128	0.275
Living condition (Living alone)	7.675	3.799	0.090	0.044	0.436	0.549	0.038	0.427
PWS	0.847	0.306	0.113	0.006	—	—	—	—
Depression	0.265	0.169	0.117	0.118	0.057	0.025	0.189	0.021
Anxiety	0.434	0.171	0.191	0.012	0.015	0.025	0.049	0.549
Stress	0.179	0.145	0.091	0.218	0.044	0.021	0.168	0.035
BMI	5.460	0.493	0.396	0.000	—	—	—	—
WBIS	—	—	—	—	0.018	0.006	0.133	0.002
Monthly income (yuan)	—	—	—	—	−0.203	0.070	−0.114	0.004
Professions (staff)	Reference							
Professions (farmer)	—	—	—	—	0.826	0.386	0.084	0.033
Professions (worker)	—	—	—	—	0.921	0.386	0.093	0.017
Professions (medical personnel)	—	—	—	—	0.025	0.121	0.009	0.835
Professions (freelance)	—	—	—	—	0.096	0.123	0.033	0.438
Professions (others)	—	—	—	—	0.162	0.177	0.037	0.361

Note: ^a^ F = 35.091, *p* = 0.000, R^2^ = 0.389, adjusted R^2^ = 0.377. ^b^ F = 15.980, *p* = 0.000, R^2^ = 0.296, adjusted R^2^ = 0.278. Abbreviation: PWS, Perceived WS questionnaire; BMI, Body Mass Index.

## Data Availability

The data presented in this study are available on request from the corresponding author. The data are not publicly available due to reasons of privacy.

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
