# Peer review of "Prevalence and Correlates of Weight Stigma among Postpartum Women in China"

_ijerph, 2022, doi:10.3390/ijerph192214692_

Round 1

Reviewer 1 Report

This paper describes an interesting survey study that explores Weight Stigma and its associated factors among post-partum women in China

Rather than calling this a convenience sample, I would specify the sampling procedure as a waiting room survey. Would it be possible to add the number of sample members approached and add a response rate? See also other methodological considerations in these papers:

Hogg, W., Johnston, S., Russell, G., Dahrouge, S., Gyorfi-Dyke, E., Kristjanssonn, E.: Conducting waiting room surveys in practice-based primary care research: A user’s guide. Can. Fam. Physician 56, 1375–1376 (2010)

Ongena, Y. P., and M. Haan. 2022. Just you wait… and fill out this survey. Discussion of the methodological aspects of waiting room surveys. Health Services and Outcomes Research Methodology.

Rookey, B. D., L. Le, M. Littlejohn, and D. A. Dillman. 2012. Understanding the resilience of mail-back survey methods: An analysis of 20years of change in response rates to national park surveys. Social Science Research 41(6):1404–1414.

In the discussion the authors write "practical constraints, the percentage of unmarried postpartum women in the sample 364 was limited". it is not clear to me what were those practical constraints; could you clarify?

Reviewer 2 Report

I found the paper interesting. I am not qualified to judge statistical methods,  but the statistics do not seem to support a statement that western standards are a cause of the desire for thinness.  I'd assume wealthy ppl and urban dwllers would have more access to int'l media, given also prominent role of english in social mobility/urban citizenship.

I'd also caution against blanket statements about 'tradition' and simplistic 'tradition vs western' schemas. The comparative analysis needs to be clarified and if authors feel small sample size is too small to judge then why include. 

Lack of attention to men in this study vs others should be explaind.

Needs to incorporate lit on body image in China. this must exist.
